# Evaluating integrated surface/subsurface permafrost thermal hydrology models in ATS (v0.88) against observations from a polygonal tundra site

Ahmad Jan[1], Ethan Coon[1], and Scott L. Painter[1]

[1]Climate Change Science Institute and Environmental Sciences Division, Oak Ridge National Laboratory, Oak Ridge, Tennessee, USA

**Correspondence:** Scott Painter (paintersl@ornl.gov)

**Abstract.** Numerical simulations are essential tools for understanding the complex hydrologic response of Arctic regions to a warming climate. However, strong coupling among thermal and hydrological processes on the surface and in the subsurface and the significant role that subtle variations in surface topography have in regulating flow direction and surface storage lead to significant uncertainties. Careful model evaluation against field observations is thus important to build confidence.

We evaluate the integrated surface/subsurface permafrost thermal hydrology models in the Advanced Terrestrial Simulator (ATS) against field observations from polygonal tundra at the Barrow Environmental Observatory. ATS couples a multiphase, three-dimensional representation of subsurface thermal hydrology with representations of overland nonisothermal flows, snow processes, and surface energy balance. We simulated thermal hydrology of a three-dimensional ice-wedge polygon with geometry that is abstracted but broadly consistent with the surface microtopography at our study site. The simulations were forced by

meteorological data and observed water table elevations in ice-wedge polygon troughs. With limited calibration of parameters appearing in the soil evaporation model, the three-year simulations agreed reasonably well with snow depth, summer water table elevations in the polygon center, and high-frequency soil temperature measurements at several depths in the trough, rim, and center of the polygon. Upscaled evaporation is in good agreement with flux tower observations. The simulations were found to be sensitive to parameters in the bare soil evaporation model, snowpack, and the lateral saturated hydraulic conduc-

tivity. Timing of fall freeze-up was found to be sensitive to initial snow density, illustrating the importance of including snow aging effects. The study provides new support for an emerging class of integrated surface/subsurface permafrost simulators.

# 1 Introduction

Permafrost soils underlie approximately one quarter ($\sim$15 million km$^2$) of the land surface in the Northern Hemisphere (Brown et al., 1997; Jorgenson et al., 2001), and store a vast amount of frozen organic carbon (Hugelius et al., 2014; Schuur et al., 2015). Warming in Arctic regions is expected to lead to permafrost thawing, as has been observed from field data during the past several decades (Lachenbruch and Marshall, 1986; Romanovsky et al., 2002; Osterkamp, 2003; Hinzman et al., 2005; Osterkamp, 2007; Wu and Zhang, 2008; Batir et al., 2017; Farquharson et al., 2019). For example, a very recent field study in the Canadian High Arctic, a cold permafrost region, reported the observed active-layer thickness (ALT, annual maximum thaw depth) already exceeds the ALT projected for 2090 under RCP 4.5 (Representative Concentration Pathways) (Farquharson et al., 2019). The thermal stability of these regions is a primary control over the fate of the stored organic matter. Since most of this organic carbon is stored in the upper 4 m of the soil (Tarnocai et al., 2009), degradation of permafrost can result in the decomposition of large carbon stocks, potentially releasing this carbon to the atmosphere (Koven et al., 2011). Warming and permafrost degradation can also contribute to hydrological changes in the northern latitudes (Osterkamp, 1983; Walvoord and Striegl, 2007; Lyon et al., 2009; Yang et al., 2010; Pachauri et al., 2014), causing substantial impact on the Arctic ecosystem. As climate models generally indicate accelerating warming in the 21st century, there is an urgent need to understand these impacts.

Process-based models are essential tools for understanding the complex hydrological environment of the Arctic. One-dimensional models of subsurface water and energy transport that incorporate freezing phenomena have a long history; comprehensive reviews are provided by (Kurylyk et al., 2014; Kurylyk and Watanabe, 2013; Walvoord and Kurylyk, 2016). Those one-dimensional models have been adapted to model the impacts of climate warming on permafrost thaw and the associated hydrological changes at regional and pan-Arctic scales (Jafarov et al., 2012; Slater and Lawrence, 2013; Koven et al., 2013; Gisnås et al., 2013; Chadburn et al., 2015; Wang et al., 2016; Guimberteau et al., 2018; Wang et al., 2019; Tao et al., 2018; Yi et al., 2019). At the smaller scales and higher spatial resolutions required to assess local impacts, processes that can be neglected at larger scales come into play creating additional modeling challenges (Painter et al., 2013). Those challenges include strong coupling among the hydrothermal processes on the surface and in the subsurface, the important role of lateral surface and subsurface flows, and in some situations the role of surface microtopography (Liljedahl et al., 2012; Jan et al., 2018a) in regulating flow direction and surface water storage.

In recent years, cryohydrogeological simulation tools capable of more detailed three-dimensional (3D) representations of subsurface processes have been developed (McKenzie et al., 2007; Rowland et al., 2011; Tan et al., 2011; Dall'Amico et al., 2011; Painter, 2011; Karra et al., 2014). Cryohydrogeological tools typically couple Richards equation for variably saturated 3D subsurface flow with 3D heat transport models using either empirical soil freezing curves (McKenzie et al., 2007; Rowland et al., 2011) or physics-based constitutive relationships (Tan et al., 2011; Dall'Amico et al., 2011; Painter, 2011; Karra et al., 2014). The physics-based constitutive relationships among temperature, liquid pressure, gas and liquid saturation indices are deduced from unfrozen water characteristic curves, capillary theory and the Clapyeron equation (Koopmans and Miller, 1966; Spaans and Baker, 1996; Painter and Karra, 2014). Notably, models with physics-based constitutive relationships have been

quite successful at reproducing laboratory freezing soil experiments, in some cases (Painter, 2011; Painter and Karra, 2014; Karra et al., 2014) without recourse to empirical impedance functions in the relative permeability model. This class of models and similar approaches implemented in proprietary flow solvers have been used to gain insights into permafrost dynamics in saturated conditions with no gas phase (Walvoord and Striegl, 2007; Bense et al., 2009; Walvoord et al., 2012; Ge et al., 2011; Bense et al., 2012; Wellman et al., 2013; Grenier et al., 2013; Kurylyk et al., 2016) and in variably saturated conditions with a dynamic unsaturated zone (Frampton et al., 2011; Sjöberg et al., 2013; Frampton et al., 2013; Kumar et al., 2016; Schuh et al., 2017; Evans and Ge, 2017; Evans et al., 2018).

Cryohydrogeologic models only represent the subsurface and must be driven by land surface boundary conditions on infiltration, evapotranspiration, and surface temperature. The typical approach in applications is to use empirical correlation to meteorological conditions to set those boundary conditions. Given the strong coupling between surface and subsurface flow systems when the ground is frozen and the key role that surface energy balance and snowpack conditions play in determining subsurface thermal conditions, the lack of a prognostic model for surface flow and surface energy balance introduces additional uncertainties when used in projections to assess hydrological impacts of climate change.

Notably, integrated surface/subsurface models of permafrost thermal hydrology have recently started to appear. The GeoTop 2.0 (Endrizzi et al., 2014) and the Advanced Terrestrial Simulation (Coon et al., 2016; Painter et al., 2016) models couple 3D cryogeohydrological subsurface models with models for overland flow; snow accumulation, redistribution, aging, and melt; and surface energy balance including turbulent and radiative fluxes and the insulating effects of the snowpack. Nitzbon et al. (2019) recently extended the thermal-only simulator Cryogrid 3 (Westermann et al., 2016) to include a simplified hydrology scheme that avoids solving the computationally demanding Richards equation. All of these models remove the requirement for imposing surface conditions and as such offer the potential for advancing understanding of permafrost thermal hydrology as an integrated surface/subsurface system (Harp et al., 2015; Atchley et al., 2016; Pan et al., 2016; Sjöberg et al., 2016; Jafarov et al., 2018; Abolt et al., 2018; Nitzbon et al., 2019).

Despite the advances in integrated thermal hydrology of permafrost, model evaluation against field observations remains a major challenge (Walvoord and Kurylyk, 2016). After successful code verification, the next question becomes how well these process-based models can reproduce the current state of the permafrost at the scale of field observations. That model evaluation against field observation is important to build confidence in process-based models. Once carefully evaluated, models can then provide insight into recent changes (such as thermokarst development and talik formation) and future evolution under different climate scenarios at watershed scales. To date, model evaluation has largely been restricted to soil temperature data (Endrizzi et al., 2014; Atchley et al., 2015; Harp et al., 2015; Sjöberg et al., 2016; Abolt et al., 2018; Nitzbon et al., 2019). Those comparisons to soil temperature measurements are an important first step in building confidence in recently developed process-rich permafrost thermal hydrological models. However, temperature data alone have been shown to be a weak constraint on model parameters (Harp et al., 2015) and do not adequately test representations of many important physical processes such as lateral water flows, advective heat transfer, wind-driven snow distribution, and microtopography-induced preferential flow paths and water storage.

In this paper, we evaluate integrated surface/subsurface permafrost thermal hydrology models implemented in the Advanced Terrestrial Simulator (ATS) v0.88 using soil temperature (Romanovsky et al., 2017; Garayshin et al., 2019), water level (Liljedahl and Wilson., 2016; Liljedahl et al., 2016), snowpack depth (Romanovsky et al., 2017), and evapotranspiration (Dengel et al., 2019; Raz-Yaseef et al., 2017) data collected over several years at the Next Generation Ecosystem Experiment-Arctic (NGEE-Arctic) study site in polygonal tundra near Utqiaġvik (formerly Barrow), Alaska. Simulations are driven by observed meteorological data (air temperature, snow precipitation, rain precipitation, wind speed, relative humidity, incoming longwave radiations, and incoming shortwave radiations) and observed water table elevations in polygon troughs. Simulated results are compared with multiyear observations of water table in the polygon center, soil temperatures at several depths (0-1.5 meters) across three microtopographic positions (rim, center, trough), evaporation, and snowpack depth in the polygon center, rim, and trough. The simulations explore the sensitivity of the results to the saturated hydraulic conductivity, snowpack representation, and the soil evaporation model. The objectives of this study are to evaluate the potential of the emerging integrated surface/subsurface thermal hydrology models as tools for advancing our understanding of permafrost dynamics, build confidence in the model representations, and identify a set of model parameters that can be used in future simulations projecting permafrost thaw and degradation.

## 2 Field site and data description

Observations for our model evaluation came from the field site of the Next Generation Ecosystem Experiment (NGEE) Arctic project (https://ngee-arctic.ornl.gov) located within the Barrow Environmental Observatory (BEO) near Utqiaġvik (formerly Barrow), Alaska (see Figure 1). The BEO is located in lowland polygonal tundra. These patterned grounds developed by repeated freezing and thawing of the ground over hundreds of thousands of years, which results in subsurface ice-wedges arranged in polygonal patterns (de Koven Leffingwell, 1919; Lachenbruch, 1962; Greene, 1963; Mackay, 1990; Jorgenson et al., 2006). Typically, ice-wedge polygons are classified into high-, intermediate-, low-, and flat-centered polygons based on microtopographic relief (Black, 1982; Oechel et al., 1995; Liljedahl et al., 2012). We used observations from a low-centered and an intermediate-centered polygon from study Area C; see Figure 1 (bottom right). More details about polygons characteristics at the NGEE Arctic field sites can be found in Kumar et al. (2016).

Meteorological data for the study area were compiled from a variety of sources by Atchley et al. (2015). Temperature and precipitation for the time period of interest are shown in supplemental material (Figure S1). The annual average air temperature is -10.8, -10.4, and -9.9 °C, and total annual average precipitation is 302, 427, 308 mm for years 2012-2014, respectively. The snow precipitation includes a 30% adjustment for undercatch (Atchley et al., 2015). We applied the undercatch adjustment to the snow precipitation uniformly in time and space. As described in Section 3.2 below, ATS then distributes incoming snow precipitation nonuniformly using a phenomenological algorithm that preferentially fills microtopographic depressions first.

NGEE-Arctic scientists conducted field campaigns to collect 1) water level data in centers and troughs of the polygons during the summers of 2012-2014 (Liljedahl et al., 2016); 2) soil temperature data at several depths (from 5 cm to 150 cm) in troughs, rims, and centers of the polygons from September 2012 to October 2015 (Garayshin et al., 2019); and 3) summer

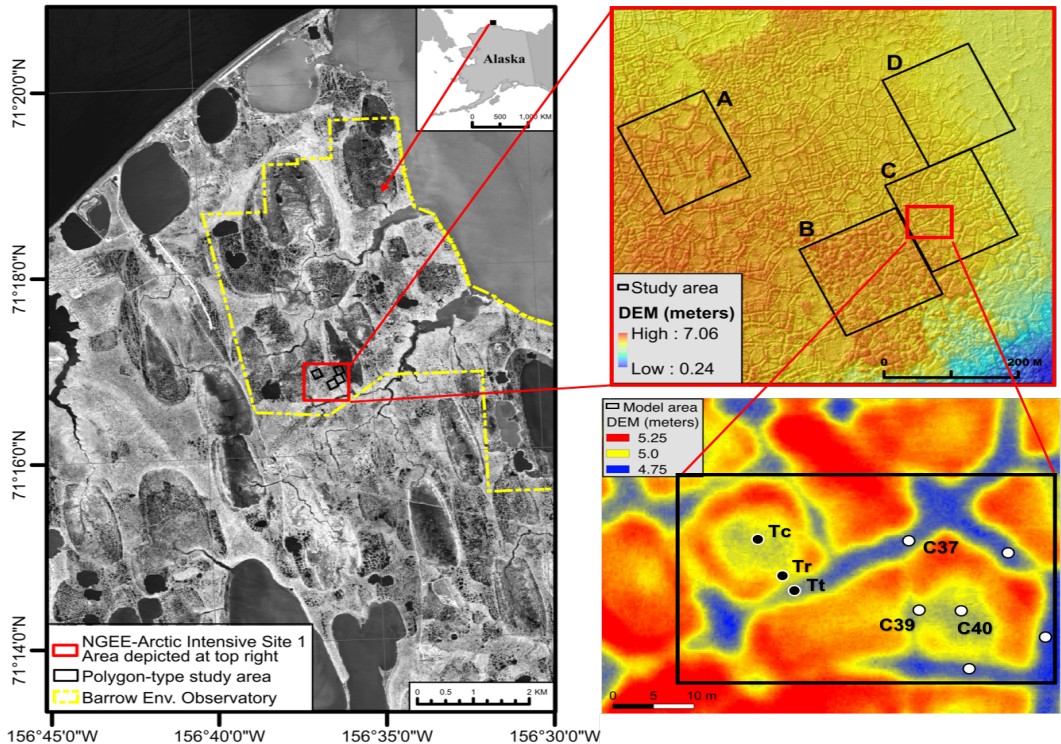

**Figure 1.** Next Generation Ecosystem Experiments – Arctic field sites at the Barrow Environmental Observatory. The model area lies in Area C outlined in black (lower right). Wells and thermistor probe locations are highlighted in white and black, respectively. Data from wells and temperature probes used for model evaluation are labelled as: C37 (trough), C39 (center), C40 (center), and vertical thermistor probes Tt (trough), Tc (center), and Tr (rim). The Digital Elevation Model (DEM) in the left two panels as derived from LiDAR measurements (Wilson and Altmann, 2017).

evapotranspiration measurements from 2012 (Raz-Yaseef et al., 2017). The water level and soil temperature measurements were recorded at 15 and 60 minutes intervals, respectively. We used data from three shallow wells (C37, C39 and C40) and from three vertical thermistor probes located in a polygon center, rim, and trough and denoted Tc, Tr, and Tt, respectively (see Figure 1, bottom right panel). The datasets are publicly available at the NGEE Arctic data portal (Romanovsky et al., 2017; Liljedahl and Wilson., 2016; Dengel et al., 2019).

## 3 Methods

### 3.1 Mesh construction

The objective of this study is to evaluate the integrated surface/subsurface models in ATS against multiple types of field observations. As described in Section 2, the temperature and water level observations are not co-located, but were obtained in two

neighboring ice wedge polygons. Rather than build faithful representations of each polygon and evaluate against temperature and water level data independently, we chose as our modeling domain a single polygon that is an abstraction of the two actual polygons. Using that abstracted geometry allows our models to be evaluated against both types of measurements simultane-

135 ously. Evaluating against multiple data types and use of a slightly abstracted but broadly representative geometry is consistent with our overarching motivation, which is to construct models that are broadly representative of the BEO site and of polygonal tundra in general.

In building the abstracted ice-wedge polygon, we imposed several constraints. For reproducing the water levels measured at wells C39 and C40, which represent polygon center locations, it is important that the surface elevation match that of the

140 measurement location. Moreover, to adequately represent overland and shallow subsurface flow, it is important to honor rim height, as Liljedahl et al. (2016) have demonstrated and we have confirmed by exploratory sensitivity simulations undertaken for this study (results not shown). We thus match the low point in the rim elevation, as that determines the spill point for surface and shallow subsurface flow between the center and trough. When comparing to soil temperature measurements it is necessary to match the surface elevation of those locations because thermal conditions are sensitive to snow depth and soil water content

(Atchley et al., 2016), which both depend on rim elevation relative to the center and trough. Based on those constraints, we constructed a 3D mesh comprising 6 equal-sized wedges (Figure 2). The surface elevation in one wedge has trough and rim elevations corresponding to that of the thermistor probes Tt and Tr, respectively. The opposite wedge matches the trough and rim surface elevation for the polygon containing the water-level observations wells. The center elevation was set by averaging the surface elevation at the two observations wells C39 and C40, which are taken to be representative of water level dynamics in

the polygon center. After those two wedges were constructed, interpolation determined the surface elevation of the 4 remaining wedges.

Given that surface elevation map, we then extruded in the vertical to create a 3D mesh. The subsurface was divided into moss, peat and mineral soil layers. Because moss is an important control on the transfer of surface energy to the permafrost (e.g., Beringer et al. (2001)), we explicitly represented it as a porous medium. That 2 cm moss layer sits atop a 8 cm layer

of peat. Regions below the peat were represented as mineral soil. The moss and peat thicknesses are broadly consistent with observations at the BEO site. For simplicity, we neglected spatial variability and modeled the moss and peat layers as having spatially constant thicknesses.

## 3.2 Model description

We used the Advanced Terrestrial Simulator (ATS) Coon et al. (2019) configured for integrated surface/subsurface permafrost

thermal hydrology (Painter et al., 2016). ATS leverages a parallel unstructured-mesh computer code for flow and transport called Amanzi (Moulton et al., 2012) and uses a multiphysics management tool known as Arcos (Coon et al., 2016) to manage coupling and data dependencies among the represented physical processes, which are encapsulated in process kernels. Arcos allows ATS to configure a complex hierarchy of mathematical models at runtime. ATS's permafrost configuration (Painter et al., 2016) solves fully coupled surface energy balance (Atchley et al., 2015), surface/subsurface thermal hydrology with

freeze/thaw dynamics, and snow distribution models. ATS represents important physical process such as lateral surface and

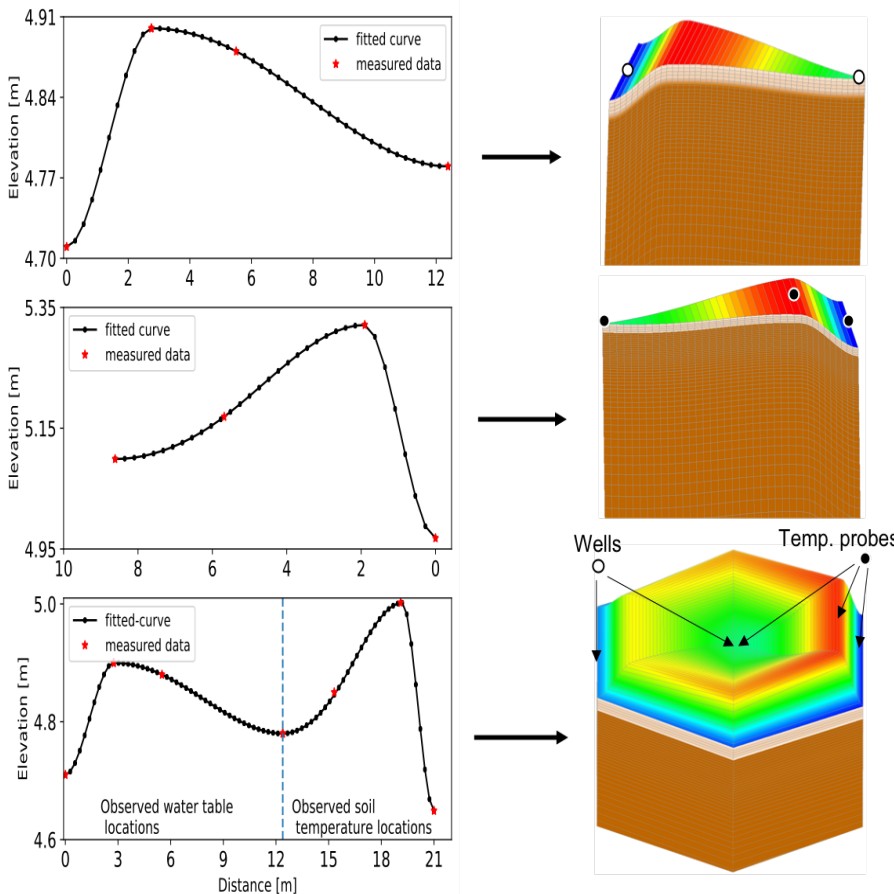

**Figure 2.** Construction of generic meshes from the observed elevations at the microtopographic locations (trough, rim, and center). Red asterisks represent measured elevation, black dots show spatial resolution (∼25 cm), and black solid lines correspond to a spline fit to the measured elevation with zero slope at the locations trough, rim and center. The bottom left combines the transects where the water level (left) and soil temperature (right) measurements were made. First a 2D surface is constructed from transect representing generic surface topography and then extended to 50 m below the surface using variable vertical resolution (lower right). Wells and thermistor probe locations are highlighted in white and black dots, respectively. The color palette in the right panel correspond to the elevation in the plots on the left side and different than the color range provided in the upper and lower right panel in Figure 1.

subsurface flows, advective heat transport, cryosuction, and coupled surface energy balance, and has been used successfully in previous studies to simulate integrated soil thermal hydrology of permafrost landscapes (Jafarov et al., 2018; Abolt et al., 2018; Sjöberg et al., 2016; Atchley et al., 2015, 2016; Schuh et al., 2017; Harp et al., 2015). Note that ATS does not require an empirical soil freezing curve to determine the unfrozen water content versus temperature. Instead, partitioning between ice, liquid, and gas is dynamically calculated from temperature and liquid pressure using the soil water characteristic curve (SWCC) in unfrozen conditions, a Clapeyron equation, and capillary theory, as described by Painter and Karra (2014). We use van Genuchten's model (van Genuchten, 1980) here. Similarly, the relative permeability in frozen or unfrozen conditions is obtained directly from the SWCC and the model of Mualem (1976) using the calculated unfrozen water content. That is, no additional empirical impedance term is introduced in the relative permeability function. The underlying soil physics models have been carefully compared (Painter, 2011; Painter and Karra, 2014; Karra et al., 2014) to published results from soil freezing experiments. ATS's surface system includes overland flow and advective heat transport with phase change of ponded water; evaporation from ponded water and bare soil; surface energy balance; a snow thermal models that accounts for aging/compaction and optionally the formation of a low conductivity depth hoar layer (enabled in our simulations); and a heuristic snow distribution model that preferentially deposits incoming snow precipitation into microtopographic depressions until those depressions are filled. The integrated surface/subsurface models have been compared successfully to soil temperature measurements (Atchley et al., 2015; Harp et al., 2015; Sjöberg et al., 2016). ATS's integrated surface/subsurface flow capabilities without freezing have been compared to other hydrological models as part of the integrated hydrologic model intercomparison project, IH-MIP2 (Kollet et al., 2017).

The ATS permafrost thermal hydrology models we are evaluating here were first implemented in ATS v0.86 and described in detail in Painter et al. (2016). The surface energy balance equation was presented by Atchley et al. (2015) and first implemented in ATS v0.83. We used ATS v0.88 here. The permafrost thermal hydrology physics and model structure were unchanged between versions 0.86 and 0.88, although there were some minor changes in input formats. ATS v0.88 has additional intermediate-scale modeling capabilities (Jan et al., 2018b, a) that are especially useful and efficient for watershed-scale simulations. The intermediate-scale variant also has dynamic topography caused by melting of massive ground ice using an algorithm proposed by Painter et al. (2013). The intermediate-scale capabilities are not exercised here.

### 3.3 Simulation description

### 3.3.1 Boundary conditions

Model evaluation was performed for years 2012-2014, due to the availability of the observation data during this period. The 3D simulations use no-flow boundary conditions in the subsurface on the vertical sides of the domain, based on an assumption of symmetry at the trough thalweg. The bottom boundary (50 m deep) is subject to -6.0 °C constant temperature (Romanovsky et al., 2010). The surface system was driven by observed air temperature, relative humidity, wind speed, rain/snow precipitation, and shortwave and longwave radiations. Snow precipitation was increased by 30% to account for undercatch (Atchley et al., 2015). The surface flow system used observed water level from C37 as a time-varying Dirichlet boundary condition for water

pressure at the ice-wedge polygon troughs. Water levels in the three unlabeled wells in Figure 1, which are also in the troughs, are almost identical to those of C37. We thus applied the C37 water levels along the entire perimeter of the 2D surface flow system. Subsurface hydraulic and thermal properties in our reference case (Table 1) are taken from Hinzman et al. (1991) and Abolt et al. (2018).

### 3.3.2 Model initialization

To reduce the computational burden in the model initialization process, we used a multistep model spin-up process. We started with an unfrozen 1D column with water table close to the surface, then froze that column from below to steady state. That 1D frozen column was then used as an initial condition for 1D integrated surface/subsurface simulations, which were forced by site meteorological data from year 2010 repeated 100 times to establish a cyclic steady state. The resulting 1D state was then mapped to the 3D model domain, which was forced by 2010 and 2011 meteorological data, completing the spinup process.

### 3.3.3 Simulations performed

A number of simulations are performed in the study, as summarized in Table 1. The basecase subsurface physical properties are provided in Table 2. We examine sensitivity of our model to three important model parameters: 1) the snow undercatch factor, 2) saturated hydraulic conductivities, and 3) the dessicated soil thickness parameter, $d_l$, which regulates evaporative flux from dry soils. Sensitivity to the representation of snow compaction/aging and its effects on thermal conductivity is also examined. We also demonstrate that the evaporation model parameters and the saturated hydraulic conductivity can be varied simultaneously in a way that leaves the water level in the polygon center approximately unchanged, indicating the existence of a null space in the parameter space. Simulations using no-flow and seepage face (free-boundary) boundary conditions in the trough are conducted to demonstrate the role of lateral fluxes between the polygon trough and center.

### 3.3.4 Efficiency metrics

We use the Nash-Sutcliffe efficiency (NSE), root mean squared error (RMSE), and coefficient of determination ($R^2$) as performance metrics. We focus on the NSE because it is scaled by variability of the observed data. It is computed as the squared difference between the observed and simulated data normalized by the variance of the observed data and subtracted from 1 (Nash and Sutcliffe, 1970). Mathematically,

$$\text{NSE} = 1 - \frac{\sum_{i=1}^{n}(\text{O}_i - \text{S}_i)^2}{\sum_{i=1}^{n}(\text{O}_i - \overline{\text{O}})^2} \tag{1}$$

Here O and S denote observed and simulated data, respectively, and $\overline{\text{O}}$ is the observed mean value. The NSE ranges between 1 and $-\infty$, where the value 1 indicates a perfect match between the observed and simulated data, and the value 0 indicates the model is only as informative as the mean of the observations.

**Table 1.** Overview of simulations conducted in the study. The parameters used in the basecase are provided in Table 2. The parameter $k$ in the intrinsic permeability and $d_l$ is bare soil evaporation model parameter. The $\times$ denotes multiplication of the basecase value by the value in the column. The values in parenthesis correspond to different simulations for sensitivity studies.

| No. | Simulation description | Corresponding figure(s) | $k\times$ | $d_l\times$ |
|---|---|---|---|---|
| 1 | Basecase | 4, 5, 6, 7 | 1 | 1 |
| 2 | Sensitivity of soil temperature to snow undercatch | 8 | 1 | 1 |
| 3 | Sensitivity of water table in the polygon center to bare soil evaporation model parameter study | 9(top) | 1 | (0.5, 2) |
| 4 | Sensitivity of water table in the polygon center to saturated hydraulic conductivity | 9(middle) | (2, 0.5) | 1 |
| 5 | Existence of null space between saturated hydraulic conductivity and $d_l$ | 9(bottom) | (2, 0.5) | (0.5, 2) |
| 6 | Lateral water fluxes between the polygon trough and center study | 10 | (2, 0.5) | 1 |
| 7 | Sensitivity of soil temperature to snow aging model study | 11 | 1 | 1 |
| 8 | Sensitivity of water table in the polygon center to hydrological boundary conditions in the trough | S1 | 1 | 1 |

**Table 2.** Subsurface physical properties used in the model. Notations $S_{uf}$ and $S_f$ denote saturated unfrozen and frozen. Note ATS takes intrinsic permeability as input instead of hydraulic conductivity because hydraulic conductivity is temperature dependent. For reference, we also include hydraulic conductivity at 25 °C.

| Parameter | Moss | Peat | Mineral |
|---|---|---|---|
| Porosity [-] | 0.90 | 0.87 | 0.56 |
| Intrinsic permeability [m$^2$] | $1.7 \times 10^{-11}$ | $9.38 \times 10^{-12}$ | $6.0 \times 10^{-13}$ |
| Hydraulic conductivity [m/s] | $18.73 \times 10^{-5}$ | $10.34 \times 10^{-5}$ | $0.67 \times 10^{-5}$ |
| Residual water content [-] | 0.0 | 0.0 | 0.2 |
| van Genuchten alpha $\alpha$ [1/m ] | $2.3 \times 10^{-3}$ | $5.1 \times 10^{-4}$ | $3.3 \times 10^{-4}$ |
| van Genuchten $m$ [-] | $2.57 \times 10^{-1}$ | $1.9 \times 10^{-1}$ | $2.48 \times 10^{-1}$ |
| Thermal conductivity ($S_{uf}$) [W/m K] | 0.75 | 0.75 | 1.1 |
| Thermal conductivity ($S_f$) [W/m K] | 1.3 | 1.3 | 1.5 |
| Thermal conductivity (dry) [W/m K] | 0.1 | 0.1 | 0.3 |
| Bare soil evaporation model parameter $d_l$ [m] | 0.1 | 0.1 | 0.1 |

## 4 Numerical results

This section is divided into two main subsections. First, we present a comparison of simulated snow depth, water table, soil temperatures, and evaporation against field observations for multiple years. Next, we demonstrate sensitivity of soil temperatures to snow aging model and to snow undercatch adjustment, and sensitivity of water table to hydraulic conductivity and bare

soil evaporation model parameter. Finally, we showcase the presence of a null space between hydraulic conductivity and bare soil evaporation model parameter.

## 4.1 Evaluation against field observations

### 235    4.1.1    Comparison to snow sensor data

Comparisons of simulated and observed snow elevation at the rim, center, and trough location are shown for the 2012-2013 and 2013-2014 winters in Figure 3. As described above, a 30% undercatch correction estimated (Atchley et al., 2015) for the 2013-2014 winter was applied. We applied the undercatch correction uniformly in space and time to the incoming precipitation. ATS then distributes the snow on the 2D surface according to the algorithm described by Painter et al. (2016). Snow depth and
snow water equivalent is dynamically tracked while accounting for compaction, sublimation, and melt.

The simulations overpredict the snow depth by about 5-10 cm in the 2013-2014 winter and underpredict snow depth by a smaller amount in the 2012-2013 winter. Given the parsimonious nature of the snow models in ATS, with no explicit representation of snow density dependence on environmental conditions, further calibration to obtain better fits to the snow depth is not a meaningful exercise. Importantly, timing of snowpack appearance and snowmelt are well represented. Significantly, distri-
bution among the center, rim, and trough locations also agrees well with the observations. These results indicate that the ATS models for snowpack dynamics and snow distribution are reasonably representative. However, it is important to note that the snow distribution model is phenomenological, specific to distributing snow in microtopography of otherwise flat landscapes, and not applicable to hilly or mountainous regions.

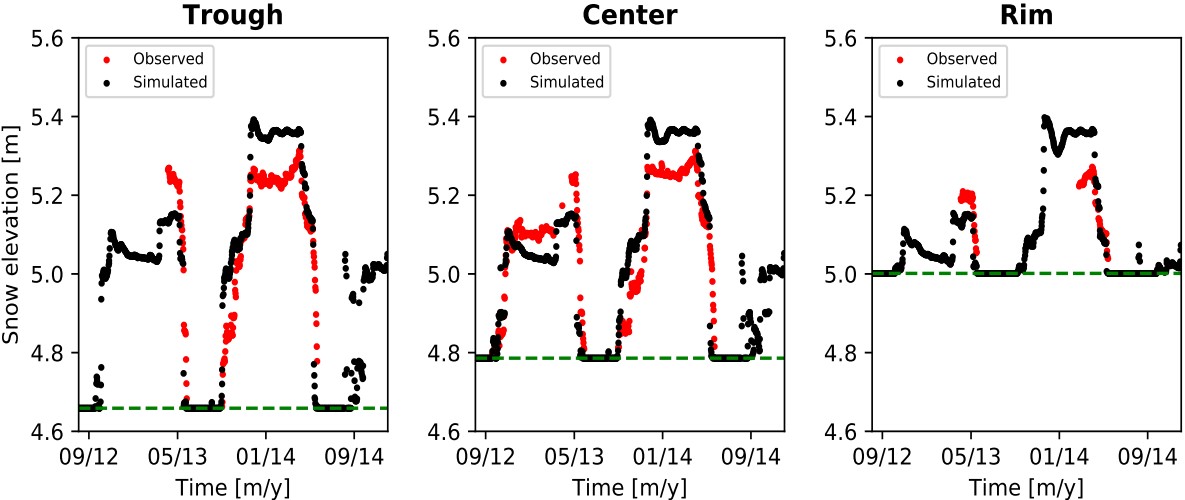

**Figure 3.** Observed and simulated snow elevation for the 2012-2013 and 2013-2014 winters at trough, center, and rim locations.

#### 4.1.2 Comparison to observed temperature data

Comparisons of simulated and observed soil temperatures are shown in Figure 4 at depths of 5 cm (near surface), 50 cm (near active layer thickness), and 150 cm (shallow permafrost). Each column corresponds to a microtopographic location, left column (trough), middle column (center), and right column (rim). Observed data is plotted in the red solid lines, simulated is the black dashed curves, and the green dashed horizontal line represents 0 °C. Simulated temperatures are in good agreement with the measured throughout the 2+ year period, with the largest discrepancy occurring in center during the winter of 2012-2013. In

general, timing of snowmelt, freeze-up, and depth of the active layer are well represented across the polygon. Note that snow cover and spatial distribution of organic matter within a polygon have great influence on the soil thermal regime due to their distinct hydrothermal properties. We have not attempted to optimize the soil organic matter thickness and only considered uniform organic matter thickness across the polygon, despite its importance in determining the temperature at depth, because our focus here is on generic simulations that can be applied without detailed site-specific characterization data.

Garayshin et al. (2019) modeled the same temperature data using a nonlinear heat-conduction model that presumes a saturated soil and neglects hydrological processes. Their simulations generally match the observed temperature at shallow depths in terms of both amplitude and phase of the seasonal signal. Their results also generally match the amplitude of the seasonal signals at depth, but show a significant phase shift at depth with the model results lagging the observations. That lag is most pronounced during the spring of 2014 where consistent error across all microtopographic positions and depths were seen.

That our simulations with a more complete representation of the thermal hydrological processes are free from those artifacts is encouraging, especially considering that we have abstracted the ice-wedge polygon geometry, microtopography, and soil structure and have not undertaken a formal calibration/parameter estimation procedure. The Nash-Sutcliffe coefficient (NSE) and root mean squared error (RMSE) of the simulated soil temperature at several depths for years 2012-2014 in comparison with the observed data are presented in Table S1 (supplemental document). The coefficient of determination ($R^2$) and model

bias are also presented. The warm (or cold) bias in the model is represented by positive (or negative) bias. The NSE in all cases is greater than 0.94, which indicates an excellent match between the measured and simulated data.

#### 4.1.3 Comparison to observed water levels

Figure 5 shows simulated water table compared to the observed water table from snowmelt to freeze-up period for years 2012-2014. Figure 5(left) shows the observed water level imposed as a Dirichlet boundary condition and the simulation result in the

center of computation grid cell adjacent to the boundary. The boundary condition acts as a run-off (outflow) or run-on (inflow) boundary condition as the observed water level in the trough drops below or rises above the simulated water level, respectively. The trough water level matches the imposed boundary conditions closely except during the 2012 summer when the water level was below the surface. We imposed a no-flow boundary condition during that period.

Figure 5(right) shows the simulated results compared with the observed in the polygon center. The observation for year 2012

is the average of the water levels for wells C39 and C40 (due to the mid-summer measurement gap at well C39), however, observation data for years 2013 and 2014 is for well C39 only. Water depths in those two closely spaced wells have only

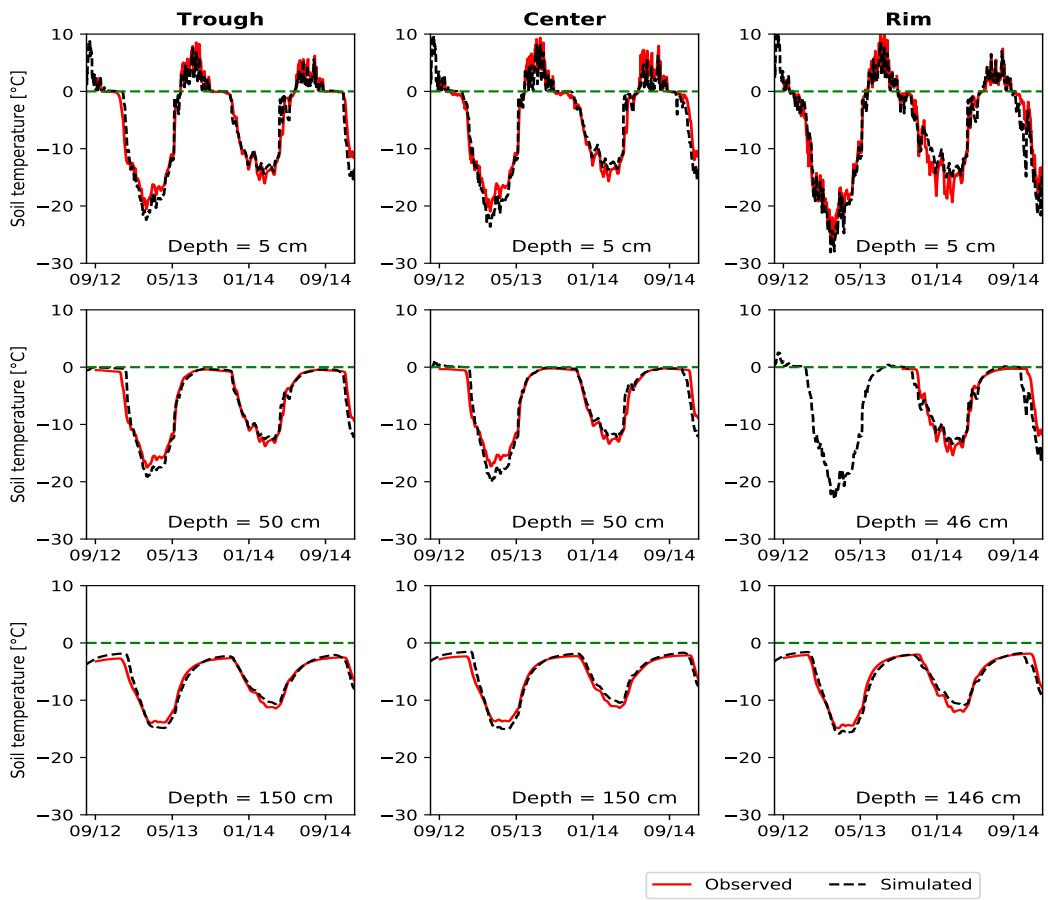

**Figure 4.** Comparison of simulated and observed soil temperatures in the trough (left column), in the center (middle column), and rim (right column) for years 2012-1014 at several depths. Rows correspond to the depth from the ground surface and display measurement and simulated soil temperatures in the organic matter layer (top row), near the depth of the active layer (middle row), and in the 150 cm deep mineral (last row).

small differences, but the surface elevations are different by 10 cm. Because the datum (surface elevation) of well C39 is more aligned with the topography, we used only well C37 in years 2013 and 2014 for comparison with the simulated water table. The uncertainty band width is 5 cm, and is based partly on the difference in the water depths for wells C39 and C40 and partly on an estimate of uncertainty in the elevation of the wells (Liljedahl and Wilson., 2016). The simulations are generally within or close to the uncertainty band around the observations except for an approximately 2 week period during the late summer of 2012, when the simulated water level is approximately 10-15 cm below the observed. That discrepancy may be caused in part by our inability to control the trough boundary condition in the dry period prior, when the trough dries out (upper left panel in 5). When troughs stay inundated throughout the summers in 2013 and 2014, simulated results show better agreement. That is late-summer drawdown is within or close to the range of uncertainty of the measured data. The NSE value for water level for years 2012-2014 is 0.56 and the RMSE is 0.07 cm, with bias of -1.3 cm.

Given the multiphysics nature of the simulations, model uncertainties associated with abstraction of the geometry, neglect of subsurface heterogeneity, potential preferential subsurface flow paths, the phenomenological nature of the bare-soil evaporation model (discussed below), and uncertainties associated with various model parameters, the agreement is reasonably good. We discuss sensitivity of the water level to parameters and model assumptions in subsubsection 4.2.

### 4.1.4 Comparison to observed evaporation data

Simulated evaporation versus time for centers, rims, and troughs can be found in Figure S2 (supplemental document). Transpiration is minor compared to evaporation at this site (Young-Robertson et al., 2018; Liljedahl et al., 2012) and is not simulated here. The simulated evaporation is not restricted by availability of water at the trough location and is largely energy limited. The same is true for the center location in the 2013 and 2014 summers. However, drying in the center location during the 2012 summer (see Figure 5) results in a dessicated soil which inhibits evaporation in that dry period. Similarly, the simulated evaporation in the rim locations is significantly lower than the trough and center in all three summers. Reduced evaporation on microtopographic highs compared with wet polygon centers and troughs is consistent with trends observed in the chamber-based evapotranspiration measurements of Raz-Yaseef et al. (2017).

Spatially resolved evapotranspiration measurements are not available at the same locations as the water level and soil temperature measurements. However, evapotranspiration measurements are available from an eddy-covariance flux tower located approximately 250 meters to the west in similar polygonal tundra (Raz-Yaseef et al., 2017). The footprint of that tower is estimated to cover approximately 2000 $m^2$ and includes wet microtopographic lows and drier microtopographic highs. Raz-Yaseef et al. (2017) estimate 37% of the footprint is standing water, 15% is wet moss, and 48% is drier microtopographic highs. To compare with the flux tower estimates, we upscale the simulated trough, center, and rim evaporation results (Figure S2 in supplemental document) using those area fractions, equating centers to wet moss, troughs to standing water, and rims to microtopographic highs. Upscaled evaporation flux obtained this way, again neglecting the contribution from transpiration, is shown versus the flux tower observations of Raz-Yaseef et al. (2017) in Figure 6. Simulated results are in good agreement with the observations for the 2013 summer. The simulated and upscaled evaporation fluxes are slightly larger than the observations

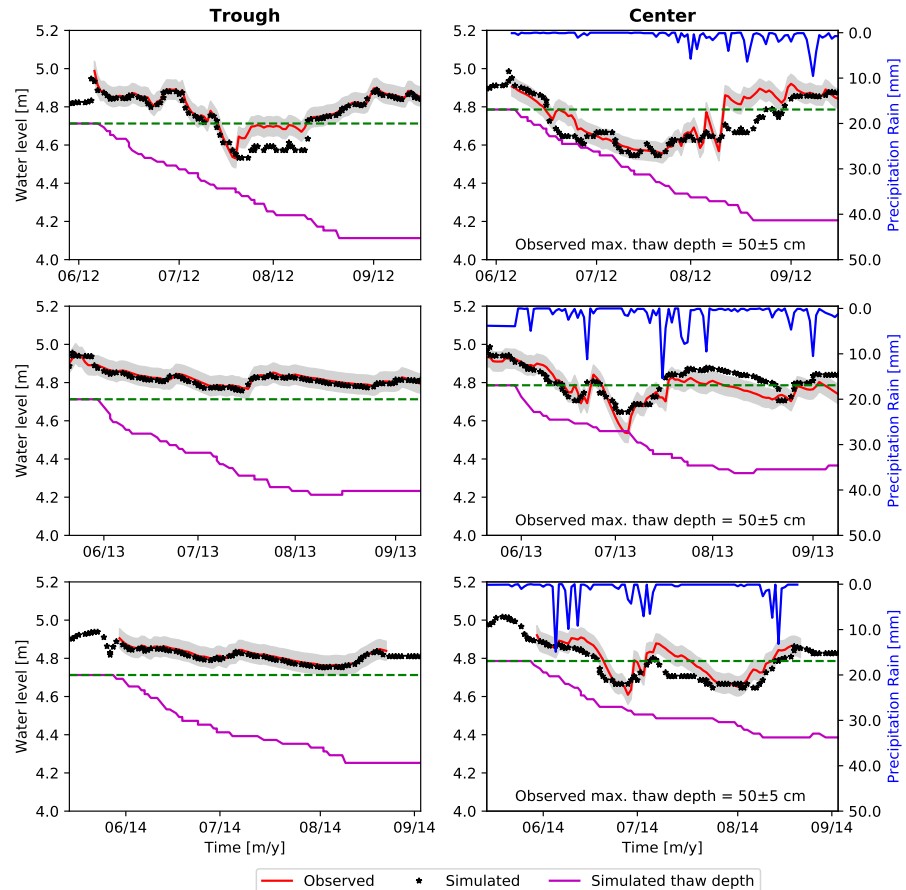

**Figure 5.** Comparison of simulated and observed water table in the trough (left column) and in the center (right column) for the summers of years 2012 to 2014. Rows correspond to different years. Blue lines (right column) is the rain precipitation.

in 2014, but reproduce the general trend. These results combined with the generally good agreement for the observed water levels provides additional support for the integrated surface/subsurface models in ATS.

## 4.2   Sensitivity analysis

### 4.2.1   Sensitivity of soil temperature to snow undercatch

Figure 7 illustrates the importance of snow undercatch adjustment. The soil temperature time-series in the trough at depths 5
320   cm, 50 cm, and 150 cm shown in red, blue, and black correspond to measured, simulated with no undercatch snow adjustment, and simulated with 30% snow undercatch adjustment, respectively. Simulated temperatures with no undercatch correction are about 2-4 degrees colder than the observed temperatures during mid-winter. The negative bias in the simulated winter temperatures is consistent across years and independent of the depth and microtopographic location. Summer temperature

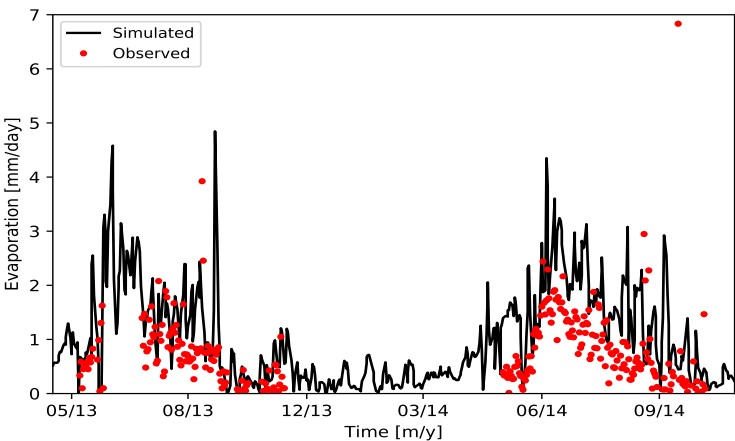

**Figure 6.** Simulated evaporation after upscaling versus observations from eddy covariance flux tower (Dengel et al., 2019; Raz-Yaseef et al., 2017).

and ALT are not affected by the snow undercatch adjustment factor and match well with the observed. The winter mismatch
between the simulated and observed temperatures is significantly improved by making a 30 % correction to the reported snow
precipitation (reference case).

### 4.2.2  Sensitivity of water table to saturated hydraulic conductivity and bare soil evaporation model parameter

Because troughs remain inundated most of the summer, flow from troughs to centers is a potentially important process for
keeping the polygon centers from drying in summer. We performed simulations in which the saturated hydraulic conductivities
of both organic matter and mineral soil were increased/decreased by a factor of 2 (see Figure 8(top)). Increasing the saturated
hydraulic conductivity enhances lateral flow from trough to center, leading to smaller drawdown than observed. Conversely,
decreasing the saturated hydraulic conductivity generally leads to drier conditions during periods of low rainfall. That the water
levels in the center is responsive to saturated hydraulic conductivity shows that lateral flow from trough to center is playing a
role in keeping the soils in the center of the polygons wet. It also demonstrates that water table measurements are informative
of the lateral saturated hydraulic conductivity as long as evapotranspiration can be independently constrained.

ATS's surface energy and water balance model includes a model for bare-soil evaporation (Sakaguchi and Zeng, 2009) that
uses a soil resistance based on vapor diffusion across a near-surface desiccated zone when the soil is dry. The maximum extent
of the desiccated zone, the parameter $d_l$ in Eq. B17 of Atchley et al. (2015), is the principal parameter in that model. Numerical
results indicate sensitivity of the water table to the bare-soil evaporation model parameter. We tested a range of values between 1
cm and 20 cm. The reference case shown in the previous section used 10 cm. Simulations show unrealistically large drawdown
of the water table during dry periods of the summer for smaller values of $d_l$. Results for $d_1 = 5$ and 20 cm are shown in Figure
8(middle). Note the case with $d_1 = 5$ cm and reference saturated hydraulic conductivity is similar to the case $d_1 = 10$ cm and
reduced saturated hydraulic conductivity shown in Figure 8(middle). That is, halving the parameter $d_1$ has a similar effect

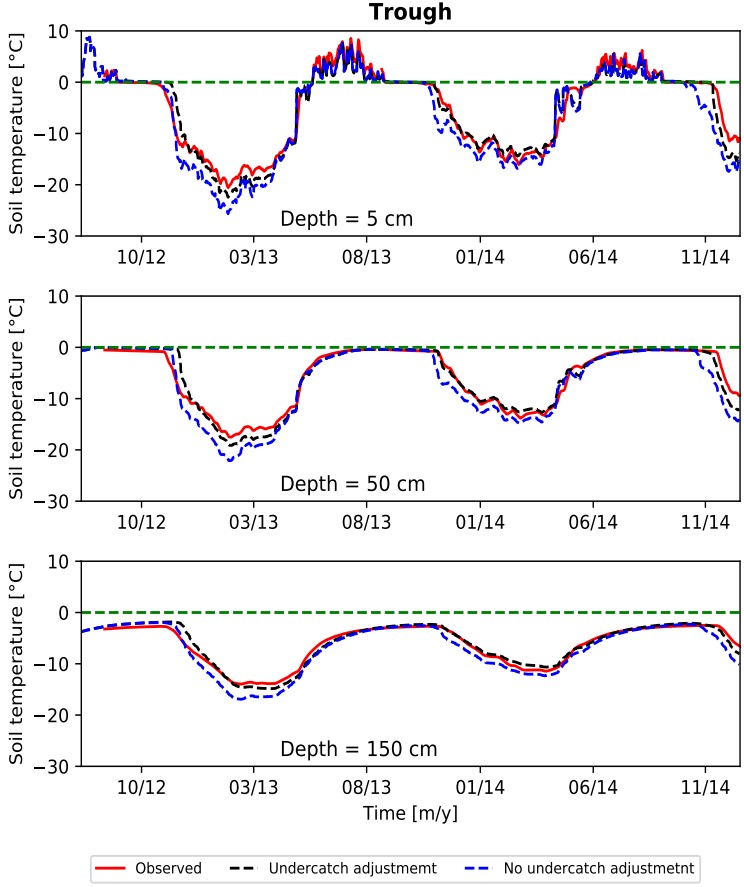

**Figure 7.** Sensitivity of the simulated soil temperatures to the snow precipitation undercatch adjustment. Smaller snowpack enhances heat escape from the soil due to the reduced insulating effect of the snowpack.

to halving the saturated hydraulic conductivity as far as drawdown during summer dry periods is concerned. That similarity indicates the existence of a null space involving the saturated hydraulic conductivities and the parameter $d_l$. In other words, these parameters can be varied simultaneously in a way that does not significantly alter the simulated water table Figure 8(bottom). However, soil temperatures show minimal-to-no sensitivity to parameter $d_l$ (results not shown here).

Simulated water fluxes between polygon center and trough through a 50 cm deep vertical slice at the right and left rims of polygon are displayed in Figure 9. In Figure 9, negative fluxes indicate inward flow (i.e., flow from trough to center). Water flow is generally from center to the trough in the early part of the summer as melt water drains through the partially thawed rim. Note that flow through the right side is small during this period because the thaw depth beneath the higher rim on that side has not reached a spill-point depth that allows water to flow through the rim. Around the end of July, flow reverses to be from trough to center and is similar in magnitude on the two sides. Increasing the hydraulic conductivity increases water flux from trough to center. This highlights the important role of lateral water fluxes between polygon center and trough.

For the purposes of model evaluation, we imposed a time-dependent water level on the trough from measured data. As a result, water is free to enter the model domain both as runoff and run-on, depending on the specified boundary condition and simulated water levels inside the model domain. Results for alternative choices of the surface water boundary condition are included in Supplemental Material (Figure S3) including spill-point boundary and closed boundaries on the surface domain. A spill-point boundary allows water out when the simulated water level reaches the spill point elevation, simulating runoff but no run-on, whereas the closed boundary eliminates both run-on and runoff. Unsurprisingly, both of the alternative boundary conditions result in poorer match to the observed water levels in the center as compared to our reference case boundary condition. In applications that seek to understand permafrost dynamics in a changing climate, surface water flows over larger domains will need to be simulated capture the dynamics of run-on and runoff, as in ATS's intermediate-scale variant (Jan et al., 2018b).

### 4.2.3 Sensitivity of soil temperature to snow aging model

The snow thermal model in ATS accounts for snow compaction/aging and the effect of that aging on thermal conductivity. New snow is introduced at density of 100 kg/m$^3$ and thermal conductivity of 0.029 W/m K. As a packet of snow ages, its density and thermal conductivity increase using the model described by Atchley et al. (2015). Sensitivity to the snow thermal model was tested by running an alternative model where new snow was introduced at the aged density and thermal conductivity. Temperature results from fall 2013 to end of 2015 at three depths with and without the snow aging model are shown versus observed soil temperature in Figure 10. Neglecting snow aging causes the ground to freeze about 1 month sooner that observed in fall of 2014 and by about two weeks in 2013. However, subsurface temperatures form the middle of winter until end of summer are unaffected by the snow model. These results show it is important to account for snow compaction/aging by introducing snow as lower density, lower thermal conductivity fresh snow, as in our reference case.

## 5 Conclusions

Individual components of recently developed integrated surface/subsurface permafrost models have been evaluated previously against laboratory measurements and field observations of temperatures. However, simultaneous evaluation against multiple types of observations is necessary to adequately test coupling between surface and subsurface systems and between thermal and hydrological processes. Those evaluations of the integrated system have been hindered by lack of co-located field observations. In this work, we took advantage of recently available multiyear, high-frequency observations of soil temperature, water levels, snow depth, and evapotranspiration to evaluate the integrated surface/subsurface thermal hydrological models implemented in the ATS code.

Because the water level and temperature data were not strictly co-located, we used an abstraction of the geometries of the two neighboring ice wedge polygons where the measurements were made. The resulting three-dimensional radially asymmetric ice-wedge polygon shares geometric features of the surface microtopography of the actual polygons that are understood to control surface and shallow subsurface flow. Using site meteorological data as forcing data and observed water table elevations in

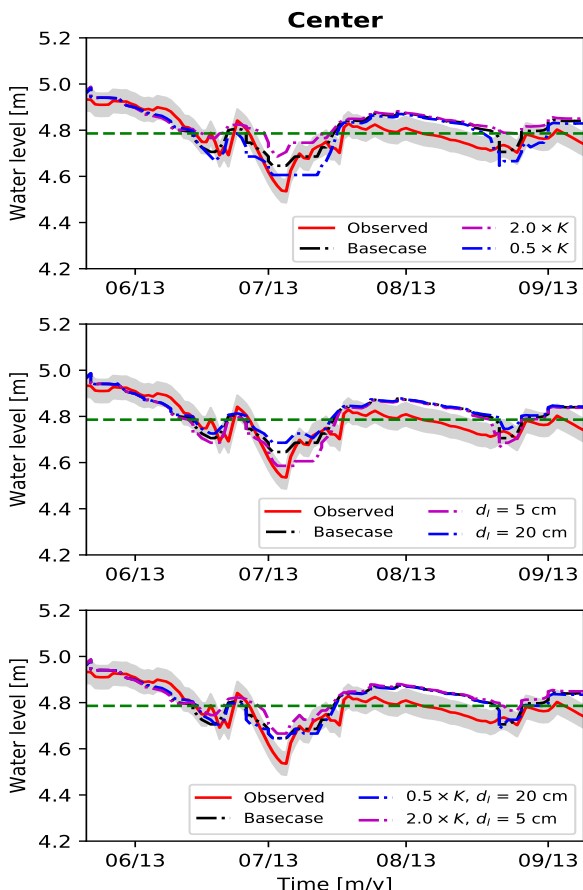

**Figure 8.** Observed water level at polygon center versus time in the summer of year 2013 showing sensitivity of the simulated water level to the bare soil evaporation model parameter (top) and saturated hydraulic conductivities (middle). Enhanced drawdown are seen in the top row when the soil evaporation parameter $d_l$ is set to smaller values. Existence of a null space between saturated hydraulic conductivity and the parameter $d_l$ is shown in the bottom row. Basecase refers to the results in Figure 5.

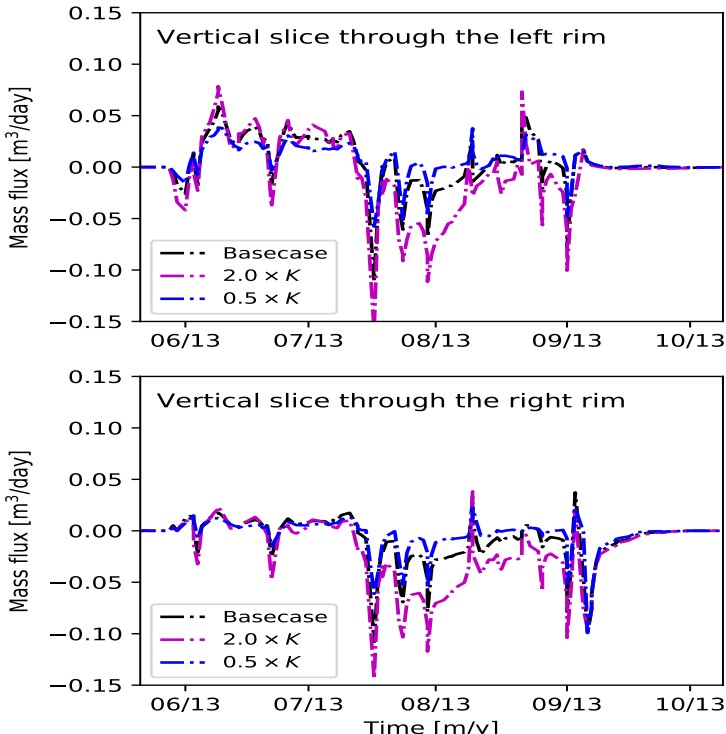

**Figure 9.** A time series of simulated lateral water fluxes between polygon center and trough through a 50 cm (the maximum thaw depth) deep vertical slice at the left and right rims of the polygon. Negative flux indicates inward flow (trough to center).

polygon troughs as a time-dependent boundary condition, we simulated water table in the polygon center and soil temperatures at several depths at three microtopographic positions (trough, rim and center). The simulations agree well with observations over three years after adjusting parameters controlling soil resistance in the bare-soil evaporation model. Other parameters were set from literature values or independently determined.

Soil temperature results were found to be sensitive to snow precipitation undercatch adjustment, consistent with the well-known thermal insulating properties of the snow pack. Timing of the fall freeze up was found to be sensitive to how the snow aging is represented. In particular, soil freezing occurred too early when snow density was assumed to be constant in time. That result demonstrates the importance of including the effects of snow aging and the formation of a depth hoar layer on thermal insulating properties of snow.

Water levels in the polygon center were found to be sensitive to the maximum extent of the soil desiccated layer, a parameter appearing in the model for soil resistance to evaporation. Water levels were also sensitive to the soil saturated hydraulic conductivity. It is important to note that the evaporation model parameters and the saturated hydraulic conductivity can be varied simultaneously in a way that leaves the water level in the polygon center approximately unchanged, indicating the existence of a null space in the parameter space. Thus, independent measurements are needed to provide additional constraints.

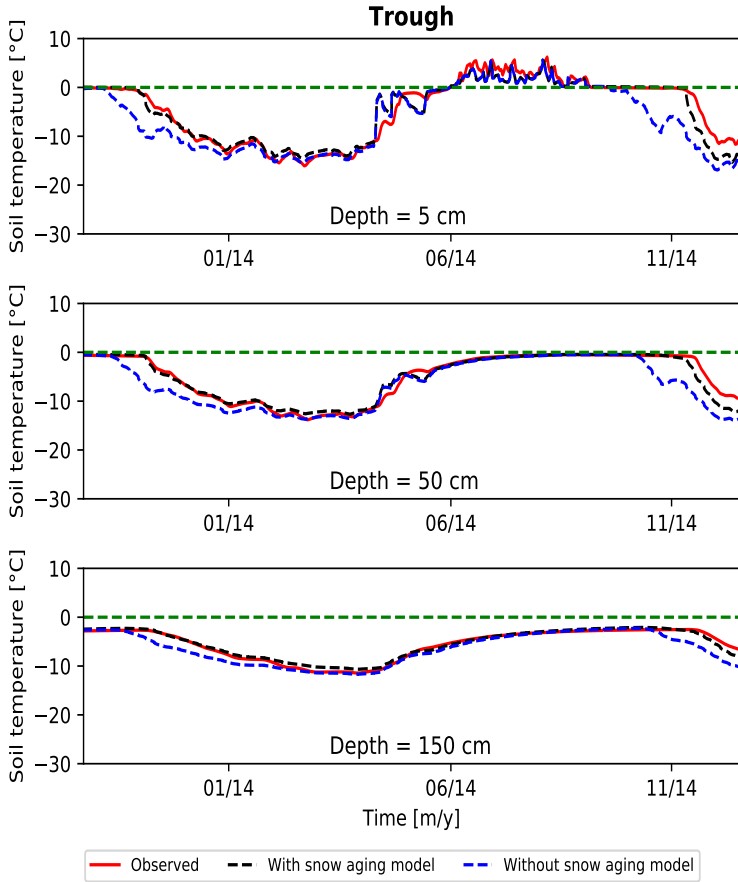

**Figure 10.** Simulated temperature versus time with and without the snow aging model compared to observed temperatures.

We used literature values to constrain the saturated hydraulic conductivity. Because those saturated hydraulic conductivity are uncertain, we took the additional step of upscaling our simulated evaporation to compare against flux tower observations, taking advantage of the fact that evaporation is dominant over transpiration at the BEO (Young-Robertson et al., 2018). Although the upscaling process has some uncertainty, the reasonably good agreement increases confidence in our representation of permafrost thermal-hydrological processes at this site. These results also demonstrate how observations of the supra-permafrost water table elevations can help constrain evapotranspiration models.

That the water levels in the polygon centers were sensitive to lateral saturated hydraulic conductivity of the subsurface underscores the role played by lateral trough-to-center subsurface flow in keeping ice wedge polygon centers from drying out in the Arctic summer.

These comparisons to multiple types of observation data represent a unique test of recently developed process-explicit models for integrated surface/subsurface permafrost thermal hydrology. The overall good match to water levels, soil temperatures, snow depths, and evaporation over the three-year observation period represents significant new support for this emerging class

of models as useful representations of polygonal tundra thermal hydrology. An obvious next step is to use this model configuration in simulations of permafrost evolution at watershed scales with large numbers of polygons represented using, for example, ATS's intermediate-scale variant (Jan et al., 2018b).

Finally, that the observations were relatively well matched by simulations that used an abstraction of the ice-wedge polygon geometry provides support for simplified geometric representations of the polygonal landscapes, which have been proposed previously (e.g. Liljedahl et al. (2012); Nitzbon et al. (2019) ). In particular, we were able to obtain good results using a regular polygon parameterized by a small number of microtopographic and soil structural parameters. Different polygon morphologies (e.g. high- versus low-center) can be represented with this parameterization by appropriate choice of those geometric quantities. In this study, we selected those parameters to represent the study site of interest. Of course, the microtopographic representation and choice of process-model parameter values are site-specific and should be evaluated for the site studied.

*Code and data availability.* The Advanced Terrestrial Simulator (ATS) Coon et al. (2019) is open source under the BSD 3-clause license, and is publicly available at https://github.com/amanzi/ats. Simulations were conducted using version 0.88. The ATS version 0.88 is permanently stored at https://doi.org/10.5281/zenodo.3727209. Forcing data, model input files, jupyter notebooks used to generate figures, meshes along with jupyter notebooks used to generate the meshes are publicly available at https://doi.org/10.5440/1545603. Data products used in the model comparisons are publicly available through the NGEE-Arctic long-term data archive https://doi.org/10.5440/1416559. The observed water level can be accessed at https://doi.org/10.5440/1183767 (Liljedahl and Wilson. (2016)), the soil temperature data at https://doi.org/10.5440/1126515 (Romanovsky et al. (2017)), and the evapotranspiration data at https://doi.org/10.5440/1362279 (Dengel et al. (2019), respectively.)

*Author contributions.* Numerical simulations were performed by AJ with guidance from SLP and ETC. All authors contributed to design of the research and to the manuscript preparation.

*Competing interests.* The authors declare that they have no conflict of interest.

*Acknowledgements.* The authors are grateful to Vladimir Romanovsky, Anna Liljedahl, Cathy Wilson, Sigrid Dengel and Margaret Torn for the BEO field data used here. The authors also thank Fengming Yuan for careful review of this manuscript. This work was supported by the Next-Generation Ecosystem Experiment—Arctic (NGEE-Arctic) project. The NGEE-Arctic project is supported by the Office of Biological and Environmental Research in the U.S. Department of Energy's Office of Science. Oak Ridge National Laboratory is managed by UT-Battelle, LLC, for DOE under contract DE-AC05-00OR22725. This research used resources of the Compute and Data Environment for Science (CADES) at the Oak Ridge National Laboratory, which is supported by the Office of Science of the U.S. Department of Energy under Contract No. DE-AC05-00OR22725.

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
