# Peer review of "Evaluating integrated surface/subsurface permafrost thermal hydrology models in ATS (v0.88) against observations from a polygonal tundra site"

_Geoscientific Model Development, 2019_

## Referee Comment (RC1) · Anonymous Referee #1 · 21 Dec 2019

The article describes various aspects of modeling of tundras ecosystem processes responses (thermal and hydrological) to atmospheric forcing. The importance of this research is obvious due to the observed and projected climate change and the ongoing Arctic amplification.

The authors present and discuss a new version of coupled three-dimentional model of heat and moisture transfer in surface and subsurface layers of soil, considering the processes of snow accumulation, redistribution and thawing, as well as turbulent and radiation flows and heat-insulating effect of snow cover ATS v0.88.

The assumptions and research methods are stated clearly. The set up and

the execution of the numerical experiments, including initial and boundary conditions setting, mesh construction, multistep model spin-up processing, and obtaining data for comparison with calculation results are described quite fully and accurately. The documentation and the source codes for the ATS model are available at https://github.com/amanzi/ats. The model is being actively used and improved, as evidenced by releases of the new versions of it.

The title matches the content of the paper, and the annotation provides a full description of the numerical experiments. The text is well structured and comprehensive. The language used is advanced and precise.

It is shown that the simulation results reproduce well the temporal dynamics of the observed parameter values, in particular, snow elevation, soil temperature, water table and evaporation. The authors also studied and assessed related research works, having provided well-selected list of references. It is also worth mentioning that ATS is a participant in projects comparing hydrological models, in particular, Kollet, S., et al. (2017) "The integrated hydrologic model intercomparison project, IH-MIP2".

Additional comments, mainly technical:

1. In Fig. 3, the right column: is the color palette similar to the upper right panel of Fig. 1? It might be a good idea to add a colorbar to it or to provide a relevant description in the figure caption for Fig. 3.

2. In Fig. 6, the authors present the thaw depths for two locations (lowland and center) for the years 2012–2014. Unfortunately, it is not quite clear from the legend and the caption if these are simulated or observed depths of thawing. If the presented depths are the results of numerical simulations, then it can be good to also show relevant observational data for comparison, or at least add maximum values for this period (50 cm depth of the zero isotherm (see Fig. 5)) that are in good agreement with modeling results.

3) In Fig. 10, the legend does not indicate observational data (red line), similar to Fig. 6.

4) For greater convenience, the tick labels could be presented in classical format: month/year or day/year (Julian Date) as, for example, in Atchley, A.L. et al (2015) "Using field observations to inform thermal hydrology models of permafrost dynamics with ATS (v0.83)".

Please also note the supplement to this comment:
https://www.geosci-model-dev-discuss.net/gmd-2019-265/gmd-2019-265-RC1-supplement.pdf

―――――――――――――――――

---

## Referee Comment (RC2) · Anonymous Referee #2 · 9 Jan 2020

In the article the authors evaluate the permafrost hydrology model ATS (v0.88, but with equivalent physics to v0.86 described in Atchley et al. (2015)), against various observations (soil temperature, water table, evaporation, snow depth) from a field site in Alaska which is characterized by ice-wedge polygons. Compared to the model description paper by Atchley et al. (2015), the article evaluates the simulations against further types of observational data, and the authors used a novel mesh geometry intended to represent various polygon morphologies by a single effective geometry. The authors furthermore conducted sensitivity analyses for a range of hydrology and snow parameterizations.

[Figure]

**General comments**

The paper is well written and addresses a relevant type of numerical model, which will be useful for an improved understanding of the thermal and hydrological dynamics of permafrost-affected terrain. Whether the presented model is also suitable for "watershed-scale projections of permafrost dynamics in a warming world", as suggested by the authors, is less obvious, as it lacks representation of surface deformation processes resulting from melting of massive ground ice.

The methodology of the study is sound, but some more details could be provided in order to assure reproducibility and to facilitate the comparison of model performance with similar types of models. Some of the assumptions should be motivated and justified in more detail (see specific comments below). While most conclusions are supported by the presented data, some conclusions would need support by the presentation of further results and/or the conduction of additional simulations (see specific comments). The title of the article is adequate and the language is mostly concise and understandable. Some sections (e.g., 3.2, 3.3, 4.5) would benefit from adding dedicated subsections. Some figures could be combined, left out, or moved to the appendix. The quality of some figures should be improved (see technical comments). The provided references are appropriate and up-to-date.

To my opinion, the article bears the potential for publication in GMD, if the authors (i) provide further justification for parts of the methodology, (ii) provide further evidence for some of the conclusions, (iii) discuss limitations of the model setup, and (iv) improve the contents and quality of the display elements.

**Specific comments**

- The assumption that the "abstracted geometry" of mixed polygon types is simultaneously representative for polygons of different types and polygonal tundra in general, is not supported by any evidence. There is evidence from other studies (e.g. Liljedahl et al. (2012)) that different polygon morphologies affect lateral hydrology in a non-linear

way. It is thus not a trivial step to assume that a "linear mixing" of different morphologies in a single radially asymmetric polygon is representative for all these morphologies at the same time. This is particularly problematic because one of the findings of the study – the strong coupling between water tables in troughs and centers – might not hold true for other types of polygons. For example, in polygons with (radially symmetric) high elevated rims, the centers and troughs would be hydrologically disconnected until the thaw depth in the rims reaches down to the elevation of the water table.

Proving that the abstracted, radially asymmetric polygon geometry is indeed representative for several polygon morphologies at a time, could instead become an objective of the study. This would, however, require complementary simulations for radially symmetric geometries of both types (high and low rims). If no further evidence for the representativeness of the "abstracted" geometry can be provided, the limitations of this setup and the validity of the conclusions should be discussed more clearly.

- Using measured water tables in polygon troughs as a forcing at the lateral boundary of the surface model domain seems rather unconventional, as such data are typically not provided by other models (as it is the case for the meteorological forcing data). In my view, the dynamic evolution of the water table throughout the thawing season is a variable a permafrost hydrology model seeks to predict based on the meteorological forcing, and the thermal and hydrological processes in the surface/subsurface system. If the elevation of the water table above the surface is, however, prescribed at the model boundary, as it is the case in the present study, the good agreement with measured water levels in the center is not very surprising, at least for low-elevated rims (see also previous comment). This procedure thus clearly limits the transferability and scalability of the approach.

Based on these simulations the authors find a good agreement between simulated and observed water tables in the polygon centers, suggesting an important role of lateral water fluxes between troughs and centers (p. 19 l. 351ff). This conclusion would become stronger if a further simulation with more simple hydrological boundary conditions for the surface (e.g., no-flow, seepage face, or a spill point at a fixed elevation) would

be conducted and included for comparison. In this respect, it might also be interesting to provide data on simulated lateral water fluxes between polygon centers and troughs (either as a time series or as net fluxes), and to assess the contribution of these fluxes to the water balance of the centers.

- The evaluation of the modeling results is mostly limited to a visual comparison between simulations and observations. For the scope of a model evaluation paper it would be desirable to provide also more quantitative measures of model performance such as RMSE, $R^2$, and/or bias. This would also facilitate the comparison with other studies that provide such numbers (e.g., Kumar et al. (2016), Abolt et al. (2018), Nitzbon et al. (2019)).

- As the active layer thickness is a key quantity for permafrost ecosystems, it would be desirable if the authors could also provide an evaluation of the temporal evolution of thaw depth, and its spatial heterogeneity between the different parts (center, rim, trough) of the polygon (provided that suitable observational data exist for BEO).

- The presented evaluation of the simulated evaporation is not very convincing. Figure 7 shows only simulation data and is thus not helpful in terms of comparison with observations. Providing the accumulated net evaporation (in [mm]) for the micro-topographic units (centers, rims, troughs), as well as providing the corresponding measured values of Raz-Yaseef et al. (2017) would be more insightful. Fig. 7 could then either be merged with Figure 8, moved to the appendix, or just left away. The time series of upscaled evaporation in Figure 8 is not suitable for a quantitative comparison between observations and simulations. It would be more insightful to provide accumulated values of net evaporation over those periods for which both measured and simulated data are available. Discussing the net evaporation together with precipitation and lateral runoff, i.e. putting it into context with the full water balance of the site, might add further relevance to the study.

- The additional simulations conducted for the sensitivity analyses are not described

in the Methods section, but rather in the Results section. The respective paragraphs should be moved to the Methods section. Making use of subsections in section 3.3 might improve readability.

- The claimed existence of a "null space", i.e. an opposing effect of saturated hydraulic conductivity and the parameter $d_l$ (p. 16, l. 309ff), is not sufficiently supported by the provided results, since only one parameter is varied at a time. Showing that a co-variation of the parameters (e.g. decreasing $d_l$ while increasing $K$) does not affect the results significantly, would strengthen this point. However, it might still be valid only for the considered polygon morphology and is not necessarily a general relation between the parameters.

- In general, the results of the sensitivity study could be explained and discussed in more detail. For example, it is a very interesting result that the initial snow density dynamics is crucial for accurately simulating accurately the duration of the zero curtain. Such insights are valuable for other modelers and could thus be elaborated more prominently.

- The limitations of the model setup should be discussed more extensively, particularly if the model is supposed to be used for projections of permafrost dynamics in a warming climate. One of these limitations is the static surface topography of the polygonal terrain, which cannot change in response to melting of massive ground ice.

- It might be considered to restructure the Results section into two parts, one for the comparison with measurements, and one for the sensitivity analyses, but each with appropriate subsections.

**Technical corrections**

- The lower right panel of Fig. 1 lacks a legend with a colorbar as it seems to be different from the one in the upper right panel.

- The information provided in Fig. 2 are not essential for the main text and could thus

be moved to the appendix. Instead, it would be sufficient to provide annual or seasonal averages for the temperature and the precipitation in the main text. It would also be interesting to provide longer-term climatological characteristics for the study area.

- The figure and axis labels in Fig. 3 should be increased and a colorbar added to the panels on the right.

- Presentation of temperature data (Figs. 2, 5, 9, 11) is much more convenient in degree Celsius instead of degree Kelvin, and would thus facilitate easier comparison with the results of other studies.

- The labels of the time axes should be presented in a more convenient format, e.g. mm/yyyy, instead of decimal years.

- The legends of Figs. 6 (right panels) and 10 are incomplete.

- Table 1 should be complemented by the values for saturated hydraulic comparison in order to facilitate comparability with other models or field sites.

- It would be nice to provide an overview over the settings of all conducted simulations, including the sensitivity analysis, e.g. in form of an additional table.

- P. 8, l. 175: not clear whether the mentioned "depth hoar" option is enabled or disabled for the presented simulations.

- P. 9, l. 194 "from literature": Provide further references if not all values are taken from Hinzmann et al. (1991).

- P. 9, l.199 "100-200 times". How many exactly? Or, why is there a range?

- P. 9, l.205-208 "As described ... sublimation, and melt.". The same information were provided already in the Methods section (where they belong) and can be left out here.

- P. 12, l. 249: "." missing

- P. 15, l.294 "Fig. 10(left)": Should be "Fig. 10(right)", correct?

**References**

All references in this review are contained in the reference list of the discussion paper.

---

## Author Comment (AC1) · 18 Feb 2020

**Response to Reviewer 1 Comments**

We thank the reviewer for the positive and constructive suggestions. Our responses are organized by reviewer comment. Italicized text are quotations from the reviews.

*It is also worth mentioning that ATS is a participant in projects comparing hydrological models, in particular, Kollet, S., et al. (2017) "The integrated hydrologic model intercomparison project, IH-MIP2.*

**Response:**

We agree with this suggestion and have included a sentence to note this at the end of the next to last paragraph in Section 3.2

*In Fig. 3, the right column: is the color palette similar to the upper right panel of Fig. 1? It might be a good idea to add a colorbar to it or to provide a relevant description in the figure caption for Fig. 3.*

**Response:** As suggested, we modified the caption and added "The color palette in the right panel correspond to the elevation in the plots on the left side and different than the color range provided in the upper and lower right panel in Figure 1."

*In Fig. 6, the authors present the thaw depths for two locations (lowland and center) for the years 2012-2014. Unfortunately, it is not quite clear from the legend and the caption if these are simulated or observed depths of thawing. If the presented depths are the results of numerical simulations, then it can be good to also show relevant observational data for comparison, or at least add maximum values for this period (50 cm depth of the zero isotherm (see Fig. 5)) that are in good agreement with modeling results.*

**Response:**

Thanks for pointing this out. We have modified Figure 6 now. The time series of the thaw depth shown is from simulations. In addition, we provide observed maximum thaw depth (text in plots) on each left side plot (center of the polygon where observed thaw depth is known). Time evolution of the thaw depth is not directly available form the observations.

*In Fig. 10, the legend does not indicate observational data (red line), similar to Fig. 6.*
**Response:** Legend has been changed to indicate observational data too.

*For greater convenience, the tick labels could be presented in classical format: month/year or day/year (Julian Date) as, for example, in Atchley, A.L. et al (2015) "Using field obser-*

*vations to inform thermal hydrology models of permafrost dynamics with ATS (v0.83).*

**Response:** In view of this suggestion, all figures have been modified (redrawn) to present x-labels in month/day format and in the plots showing temperature the unit Kelvin [K] is changed to CelsiusItalicized text are quotations from the reviews. (C).

**Response to Reviewer 2 Comments**

We are grateful to the reviewer for the detailed and constructive suggestions. We have revised the manuscript significantly to address most of these and are confident the paper has improved as a result.

Our responses are organized by comment. Italicized text are quotations from the reviews.

*The assumption that the abstracted geometry of mixed polygon types is simultaneously representative for polygons of different types and polygonal tundra in general, is not supported by any evidence. There is evidence from other studies (e.g. Liljedahl et al. (2012)) that different polygon morphologies affect lateral hydrology in a non-linear way. It is thus not a trivial step to assume that a linear mixing of different morphologies in a single radially asymmetric polygon is representative for all these morphologies at the same time. This is particularly problematic because one of the findings of the study - the strong coupling between water tables in troughs and centers might not hold true for other types of polygons. For example, in polygons with (radially symmetric) high elevated rims, the centers and troughs would be hydrologically disconnected until the thaw depth in the rims reaches down to the elevation of the water table. Proving that the abstracted, radially asymmetric polygon geometry is indeed representative for several polygon morphologies at a time, could instead become an objective of the study. This would, however, require complementary simulations for radially symmetric geometries of both types (high and low rims). If no further evidence for the representativeness of the abstracted geometry can be provided, the limitations of this setup and the validity of the conclusions should be discussed more clearly.*

**Response:** In our comparison to measured data, we used an abstracted geometry to represent two adjacent polygons where the measurements were made. These polygons have similar morphology with well-defined troughs and rims. We are not mixing polygon types, make no claim that this representation is "simultaneously representative of polygons of different types", and agree with the reviewer that such an abstraction that mixes types would need careful evaluation and is likely to be unsuccessful as a modeling strategy. It is important to note that ATS is a spatially explicit code and that large numbers of ice wedge polygons would be represented in typical applications that model over larger areas. In this model evaluation paper, we focus on small scales that were intensively monitored.

In the original manuscript, we provided physics-based justifications for our abstraction of the modeling domain, with an eye toward generalization and a wariness of overfitting due to the uniqueness-of-place challenge. We buttressed those arguments in the revised manuscript. The revised manuscript now reads "In building the abstracted ice-wedge polygon, we imposed

several constraints. For reproducing the water levels measured at wells C39 and C40, which represent polygon center locations, it is important that the surface elevation match that of the measurement location. Moreover, to adequately represent overland and shallow subsurface flow, it is important to honor rim height, as Liljedahl et al., 2012 have demonstrated. We thus match the low point in the rim elevation, as that determines the spill point for surface and shallow subsurface flow between the center and trough. When comparing to soil temperature measurements it is necessary to match the surface elevation of those locations because thermal conditions are sensitive to snow depth and soil water content (Atchley et al., 2016), which both depend on rim elevation relative to the center and trough."

Ultimately, however, the success of our abstraction can only be judged after the fact: how successful was it in reproducing observations? Readers are free to evaluate the results, which are shown in Figures 3 to 6 for multiple years and multiple types of measurements. Although there is always room for improvement, we believe that given the complexity of these systems, this is a successful comparison that provides confidence in this emerging class of multiphysics models.

In addition, we revised the final paragraphs to clarify that we are not advocating for a single polygon geometry as representative of the entire landscape. The relevant text reads:

"These comparisons to multiple types of observation data represent a unique test of recently developed process-explicit models for integrated surface/subsurface permafrost thermal hydrology. The overall good match to water levels, soil temperatures, snow depths, and evaporation over the three-year observation period represents significant new support for this emerging class of models as useful representations of polygonal tundra thermal hydrology. An obvious next step is to use this model configuration in simulations of permafrost evolution at watershed scales with large numbers of polygons represented using, for example, ATS's intermediate-scale variant (Jan et al. 2018).

Finally, that the observations were relatively well matched by simulations that used an abstraction of the ice-wedge polygon geometry provides support for simplified geometric representations of the polygonal landscapes, which have been proposed previously (e.g. Liljedahl et al. 2012; Nitzbon et al. 2019). In particular, we were able to obtain good results using a regular polygon parameterized by a small number of microtopographic and soil structural parameters. Different polygon morphologies (e.g. high- versus low-center) can be represented with this parameterization by appropriate choice of those geometric quantities. In this study, we selected those parameters to represent the study site of interest. Of course, the microtopographic representation and choice of process-model parameter values are site-specific and should be evaluated for the site studied. "

*Using measured water tables in polygon troughs as a forcing at the lateral boundary of the surface model domain seems rather unconventional, as such data are typically not provided by other models (as it is the case for the meteorological forcing data). In my view, the dynamic evolution of the water table throughout the thawing season is a variable a permafrost hydrology model seeks to predict based on the meteorological forcing, and the thermal and hydrological processes in the surface/subsurface system. If the elevation of the water table above the surface is, however, prescribed at the model boundary, as it is the case in the present study, the good agreement with measured water levels in the center is not very surprising, at least for*

*low-elevated rims (see also previous comment). This procedure thus clearly limits the transferability and scalability of the approach.*

**Response:** It is important to acknowledge the scope and goals of this study. We are taking advantage of measurements from a well-characterized site to evaluate process representations and process couplings in our process-explicit multiphysics code, consistent with the scope of a model evaluation paper in GMD. Our choice of surface water boundary condition at the polygon trough is appropriate for this model evaluation study as it eliminates uncertainty in runoff/run-on allowing us to focus on the processes of interest here. Of course trough water levels response to meteorological forcing and are of interest in a many modeling applications, but that is a completely different study suitable for a different journal.

We disagree with the assertion that a good match to the center water table is to be expected. Indeed we show in the manuscript that water table in the polygon center is sensitive to subsurface saturated hydraulic conductivity and to how bare soil evaporation is represented. Poor choice of those parameters results in a poor match to the center water table.

*Based on these simulations the authors find a good agreement between simulated and observed water tables in the polygon centers, suggesting an important role of lateral water fluxes between troughs and centers (p. 19 l. 351ff). This conclusion would become stronger if a further simulation with more simple hydrological boundary conditions for the surface (e.g., no-flow, seepage face, or a spill point at a fixed elevation) would be conducted and included for comparison. In this respect, it might also be interesting to provide data on simulated lateral water fluxes between polygon centers and troughs (either as a time series or as net fluxes), and to assess the contribution of these fluxes to the water balance of the centers.*

**Response:** We agree and have added a new figure (Figure 10 in the revised version) to quantify the lateral fluxes between polygon center and trough and include a discussion of those fluxes in the revised text. As expected, the fluxes change from center-to-trough during snowmelt to trough-to-center during dry periods. The results correspond to the basecase and 0.5 times permeability and 2 times permeability. We also provide in supplemental material (Figure S3) results from different boundary conditions (no flow and spill point). Although those BCs are not useful for our model evaluation, they do demonstrate that run-off is important during snowmelt and run-on to the polygon is important during dry summer periods. In addition to the new figure, we include the following new text at the end of Section 4.2.2:

"Simulated water fluxes between polygon center and trough through a 50 cm deep vertical slice at the right and left rims of polygon are displayed in Figure 10. In Figure 10, negative fluxes indicate inward flow (i.e., flow from trough to center). Water flow is generally from center to the trough in the early part of the summer as melt water drains through the partially thawed rim. Note that flow through the right side is small during this period because the thaw depth beneath the higher rim on that side has not reached a spill-point depth that allows water to flow through the rim. Around the end of July, flow reverses to be from trough to center and is similar in magnitude on the two sides. Increasing the hydraulic conductivity increases water flux from trough to center. This highlights the important role of lateral water fluxes between polygon center and trough.

For the purposes of model evaluation, we imposed a time-dependent water level on the trough from measured data. As a result, water is free to enter the model domain both as runoff and run-on, depending on the specified boundary condition and simulated water levels inside the model domain. Results for alternative choices of the surface water boundary condition are included in Supplemental Material (Figure S3) including spill-point boundary and closed boundaries on the surface domain. A spill-point boundary allows water out when the simulated water level reaches the spill point elevation, simulating runoff but no run-on, whereas the closed boundary eliminates both run-on and runoff. Unsurprisingly, both of the alternative boundary conditions result in poorer match to the observed water levels in the center as compared to our reference case boundary condition. In applications that seek to understand permafrost dynamics in a changing climate, surface water flows over larger domains will need to be simulated capture the dynamics of run-on and runoff, as in ATS's intermediate-scale variant Jan2018intermediate. "

*The evaluation of the modeling results is mostly limited to a visual comparison between simulations and observations. For the scope of a model evaluation paper it would be desirable to provide also more quantitative measures of model performance such as RMSE, R2 , and/or bias. This would also facilitate the comparison with other studies that provide such numbers (e.g., Kumar et al. (2016), Abolt et al. (2018), Nitzbon et al. (2019)).*

**Response:** In the revised, we include the Nash-Sutcliffe model efficiency (NSE) and RSME as performance metrics (Table S1 in supplemental material). We also revised the manuscript main text to introduce NSE as our performance metric in new section 3.3 and to provide that result in the appropriate sections. We prefer NSE in this context because it is scaled by variability.

*As the active layer thickness is a key quantity for permafrost ecosystems, it would be desirable if the authors could also provide an evaluation of the temporal evolution of thaw depth, and its spatial heterogeneity between the different parts (center, rim, trough) of the polygon (provided that suitable observational data exist for BEO).*

**Response:** Observations are not available for time evolution of the thaw depth, but we do have observed maximum thaw depth. As suggested by reviewer one also, we revised Figure 6 to include the observed maximum thaw depth (text on plots) on each left side plot.

*The presented evaluation of the simulated evaporation is not very convincing. Figure 7 shows only simulation data and is thus not helpful in terms of comparison with observations. Providing the accumulated net evaporation (in [mm]) for the micro-topographic units (centers, rims, troughs), as well as providing the corresponding measured values of Raz-Yaseef et al. (2017) would be more insightful. Fig. 7 could then either be merged with Figure 8, moved to the appendix, or just left away. The time series of upscaled evaporation in Figure 8 is not suitable for a quantitative comparison between observations and simulations. It would be more insightful to provide accumulated values of net evaporation over those periods for which both measured and simulated data are available. Discussing the net evaporation together with*

*precipitation and lateral runoff, i.e. putting it into context with the full water balance of the site, might add further relevance to the study.*

**Response:** As we described in the original manuscript, measured ET fluxes are from an eddy covariance system with flux footprint of a few hundred meters, which averages over rims, centers, and troughs. ET fluxes resolved by rim, center, and trough are not available for comparison. We believe Figure 7 (in the original version) is useful to understand that modeled evaporative flux varies by microtopographic position. Thus we keep the figure but have moved it to supplemental material as the reviewer suggested. We evaluated the option of plotting cumulative flux, but feel it is less informative than the original Figure 8, which more clearly shows seasonal variation as well as modeled evaporation when the flux tower is not operating. We are sympathetic to the suggestion of discussing the site water balance. To that end, importance of run-on and run-off can be assessed from the new results in Figure S3. However, these will be different for different positions in the catchment. Given the goals of this model evaluation paper and the small size of the modeling domain, we feel a detailed discussion of water balance would be distraction and more appropriate for another type of journal.

*The additional simulations conducted for the sensitivity analyses are not described in the Methods section, but rather in the Results section. The respective paragraphs should be moved to the Methods section. Making use of subsections in section 3.3 might improve readability.*

**Response:** The paragraph has been moved to the Methods section, and Methods section is divided into subsections to improve readability as suggested.

*The claimed existence of a null space, i.e. an opposing effect of saturated hydraulic conductivity and the parameter dl (p. 16, l. 309ff), is not sufficiently supported by the provided results, since only one parameter is varied at a time. Showing that a covariation of the parameters (e.g. decreasing dl while increasing K) does not affect the results significantly, would strengthen this point. However, it might still be valid only for the considered polygon morphology and is not necessarily a general relation between the parameters.*

**Response:** We agree with this suggestion and have modified Figure 10 (Figure 8 in the revised) to demonstrate the existence of the null space. In particular, dl and K are varied simultaneously in the new Figure 8c. The results are not significantly different from the base case, thus demonstrating the existence of a null space

*In general, the results of the sensitivity study could be explained and discussed in more detail. For example, it is a very interesting result that the initial snow density dynamics is crucial for accurately simulating accurately the duration of the zero curtain. Such insights are valuable for other modelers and could thus be elaborated more prominently.*

**Response:** The original manuscript does highlight this result in the conclusion section. In addition, we added the following: "That result demonstrates the importance of including snow-aging effects and the formation of a depth hoar layer." The following was added to the

abstract "Timing of fall freeze-up was found to be sensitive to initial snow density, illustrating the importance of including snow aging effects." Given the appropriate scope for a GMD model evaluation paper, we are constrained from going too deep into discussion of geoscience (see, e.g., GMD editors' guidance to authors).

*The limitations of the model setup should be discussed more extensively, particularly if the model is supposed to be used for projections of permafrost dynamics in a warming climate. One of these limitations is the static surface topography of the polygonal terrain, which cannot change in response to melting of massive ground ice.*

**Response:** Dynamic topography is not relevant for this model evaluation, as significant subsidence did not occur in the 3-year study period. However, ATS does include the capability to simulate subsidence caused by melting of massive ground ice. In response to this comment, we changed the last sentence in Section 3.2 to the following: "ATS v0.88 has additional intermediate-scale modeling capabilities (Jan et al., 2018a, 2018b) that are especially useful and efficient for watershed-scale simulations. The intermediate-scale variant also has dynamic topography caused by melting of massive ground ice using an algorithm proposed by Painter et al., 2013. The intermediate-scale capabilities are not exercised here."

*It might be considered to restructure the Results section into two parts, one for the comparison with measurements, and one for the sensitivity analyses, but each with appropriate subsections.*

**Response:** We have taken this suggestions in the revised manuscript.

*The lower right panel of Fig. 1 lacks a legend with a colorbar as it seems to be different from the one in the upper right panel.*

**Response:** Lower right panel has been updated to include a legend with a color bar.

*In Fig. 3, the right column: is the color palette similar to the upper right panel of Fig. 1? It might be a good idea to add a colorbar to it or to provide a relevant description in the figure caption for Fig. 3.*

**Response:** We modified the caption and added "The color palette in the right panel correspond to the elevation in the plots on the left side and different than the color range provided in the upper right column in Figure 1."

*The information provided in Fig. 2 are not essential for the main text and could thus be moved to the appendix. Instead, it would be sufficient to provide annual or seasonal averages for the temperature and the precipitation in the main text. It would also be interesting to provide longer-term climatological characteristics for the study area.*

**Response:** We have moved the figure to the supplemental material and include annual averages in the main text, as suggested

*The figure and axis labels in Fig. 3 should be increased and a colorbar added to the panels on the right.*

**Response:** The figure and axes labels are increased as suggested. Also, text has been added to the description of the figure regarding color palette. The surface elevation (colors) correspond to the elevation provided in the left panel.

*Presentation of temperature data (Figs. 2, 5, 9, 11) is much more convenient in degree Celsius instead of degree Kelvin, and would thus facilitate easier comparison with the results of other studies.*

**Response:** In view of this suggestion, all figures have been modified (redrawn) to present x-labels in month/day format and in the plots showing temperature the unit Kelvin [K] is changed to Celsius [C].

*The labels of the time axes should be presented in a more convenient format, e.g. mm/yyyy, instead of decimal years.*

**Response:** Agreed. See above comment.

*The legends of Figs. 6 (right panels) and 10 are incomplete.*

**Response:** All figures have been modified (replotted) and this has been corrected.

*Table 1 should be complemented by the values for saturated hydraulic comparison in order to facilitate comparability with other models or field sites.*

**Response:** Because saturated hydraulic conductivity is temperature dependent through the viscosity, absolute permeability is the more convenient parameter for nonisothermal applications. For readers who prefer saturated hydraulic conductivity, we include the saturated hydraulic conductivity at 25 C in the table .

*- It would be nice to provide an overview over the settings of all conducted simulations, including the sensitivity analysis, e.g. in form of an additional table.*

**Response:** We added a new Table 1 to summarize all simulations. The previous Table 1 is now Table 2.

*P. 8, l. 175: not clear whether the mentioned depth hoar option is enabled or disabled for the presented simulations.*

**Response:** It has been made clear in the same line that the depth hoar option is turned on in our simulations.

*P. 9, l. 194 from literature: Provide further references if not all values are taken from Hinzmann et al. (1991).*

**Response:** Text has been modified and one more reference has been added.

*P. 9, l.199 100-200 times. How many exactly? Or, why is there a range?*

**Response:** It should be 100. Corrected.

*P. 9, l.205-208 As described … sublimation, and melt.. The same information were provided already in the Methods section (where they belong) and can be left out here.*

**Response:** We believe it is useful for readability purposes to keep this sentence here.

*P. 12, l. 249: ”.” missing*

**Response:** Corrected

*P. 15, l.294 Fig. 10(left): Should be ”Fig. 10(right)”, correct?*

**Response:** Corrected

---

## Referee Report (RR1)

Review

**Evaluating integrated surface/subsurface permafrost thermal hydrology models in ATS (v0.88) against observations from a polygonal tundra site**

Having taken into consideration all the comments from the reviewers, the article was edited and finalized. The changes have been introduced to the abstract of the article, to the sections 2 to 4 and to the list of references, considering all the reviewers remarks. In particular, sections 3 and 4 of the text manuscript were significantly revised, and quantitative metrics (RMSE, NSE) were added to determine the model efficiency (Table S1 supplemental materials). Moreover, Table 1 was added to the text manuscript in order to specify the algorithms of numerical simulations presented in the article. In addition, the required supplemental materials were added. All the editorial remarks were taken into consideration; the ticks labels of all the figures were changed into month/day format, and the temperature units were changed from Kelvin to Celsius, as was advised.

To sum up, the text of the article is well structured and comprehensive. The language used is fluent and precise. The amount and quality of supplementary material is appropriate.